# Trends and Disparities of Energy Intake and Macronutrient Composition in China: A Series of National Surveys, 1982–2012

**DOI:** 10.3390/nu12082168

**Published:** 2020-07-22

**Authors:** Zhenni Zhu, Xiaoguang Yang, Yuehui Fang, Jian Zhang, Zhenyu Yang, Zhu Wang, Ailing Liu, Li He, Jing Sun, Yiyao Lian, Gangqiang Ding, Yuna He

**Affiliations:** 1National Institute for Nutrition and Health, Chinese Center for Disease Control and Prevention, 27 Nanwei Road, Xicheng District, Beijing 100050, China; zhuzhenni@scdc.sh.cn (Z.Z.); yangxg@ninh.chinacdc.cn (X.Y.); fangyh@ninh.chinacdc.cn (Y.F.); zhangjian@ninh.chinacdc.cn (J.Z.); yangzy@ninh.chinacdc.cn (Z.Y.); wangzhu@ninh.chinacdc.cn (Z.W.); liual@ninh.chinacdc.cn (A.L.); heli@ninh.chinacdc.cn (L.H.); sunjing@ninh.chinacdc.cn (J.S.); lianyy@ninh.chinacdc.cn (Y.L.); dinggq@chinacdc.cn (G.D.); 2Division of Health Risk Factors Monitoring and Control, Shanghai Municipal Center for Disease Control and Prevention, 1380 West Zhongshan Road, Shanghai 20036, China

**Keywords:** dietary energy, macronutrient composition, trend, disparity, subpopulation

## Abstract

Background: China’s diet transition might offer guidance to undeveloped countries on the way to prosperity. This report describes the trends and disparities in energy and macronutrient composition among Chinese adults, and between subpopulations. Methods: Data for the current study were obtained from the 1982, 1992, 2002, and 2010–2012 China National Nutrition Survey (CNNS) rounds, which were nationally representative cross-sectional surveys. We applied 24-h dietary recall and food weighing to assess dietary intake. Results: There were 204,877 participants aged 20 years or older included in the current analysis. From 1982 to 2012, the estimated energy intake declined from 2614.7 kcal to 2063.9 kcal. The trend in the estimated percentage of energy intake from fat showed a spike. It increased from 16.3% to 33.1% (1992 vs. 1982 difference, 7.6%; 95% CI 7.4% to 7.7%; 2002 vs. 1992 difference, 7.7%; 95% CI 7.6% to 7.9%; 2012 vs. 2002 difference, 1.6%; 95% CI 1.4% to 1.7%; *p* < 0.01 for trend). The trends coincided in all the subgroups (all *p* < 0.01 for trend) except for the subgroup of those educated over 15 years, whose percentage of energy intake from fat declined from 37.4% to 36.6% (2012 vs. 2002 difference, −0.8%; 95% CI −1.6% to 0.0%). The estimated percentage of energy intake from carbohydrates declined from 74.0% to 55.0%. The ranges of the estimated percentage of energy intake from fat, within population subgroups stratified by education level, area and Gross national product (GNP) level, were narrowed. Conclusions: Quick improvements in society and the economy effectively curbed undernutrition, but easily triggered overnutrition. Disparities persistently existed between different subpopulations, while the gaps would narrow if comprehensive efforts were made. Education might be a promising way to prevent overnutrition during prosperous progress. The low-social profile populations require specific interventions so as to avoid further disease burdens.

## 1. Introduction

China has been one of the fastest-growing countries over the past three decades. It implemented major social and economic reforms in 1979, and achieved tremendous economic and agricultural productivity improvement [1]. Changes in the economy, food supply and nutrition-related policies can affect diet quality at the population level. The mass Chinese population consume diets that have developed from scarcity to prosperity within only one decade or two, but this has cost a lot with regards to health outcomes, in that the burden of diet-related non-communicable disease has increased [2]. Malnutrition covers two broad groups of conditions: undernutrition and overnutrition [3]. Many developing countries work on the problem of undernutrition, while overnutrition soon emerges [4]. China’s diet transition might give guidance for the developing countries on the way to prosperity. We try to take a close look at this dietary transition during this extraordinary time in China. This report describes data from four rounds of the China National Nutrition Survey (CNNS), from 1982 to 2012. We examined the trends in energy and macronutrient composition among the Chinese population, and we also determined the disparities in dietary quality between subpopulations in terms of area, education level and economic background.

## 2. Materials and Methods

### 2.1. Study Population and Sampling

Data for the current study were obtained from the 1982, 1992, 2002 and 2010–2012 CNNS rounds, which were nationally representative cross-sectional surveys conducted by the Chinese Center for Disease Control and Prevention in order to assess the health and nutrition of the Chinese population [5]. The design, sampling and dietary data collection methods of each round were homogeneous. The survey design and methods have been presented in detail previously [5]. A stratified and multistage cluster randomized sampling method was applied. There were initially 238,124, 100,201, 247,464 and 188,622 participants recruited in the surveys from 1982 to 2012, respectively. The response rate was 87.9% in 2002 and 76.5% in 2010–2012. Response rates were not recorded in the 1982 and 1992 surveys. All CNNS rounds collected identical data from household and dietary interviews, body measurements and laboratory tests. Some participants were selected to participate in certain survey items, while others participated in another. For this analysis, we restricted the study sample to adults aged 20 years or older with dietary intake data.

Education level was defined as years the participant had received education. Area was defined as urban or rural because China is a two-class society with rural–urban distinctions in many aspects. The urban sector has gained more benefits from social and economic reforms than the rural sector has. Life style and dietary pattern were distinguishing between the two sectors. Gross national product (GNP) level was classified by provincial level according to the GNP quartiles across provinces. In 1982, the first to fourth classes were classified as ≥284, (244, 284), (194, 244) and <194, respectively, in USD; in 1992, the first to fourth classes were classified as USD ≥482, (354, 482), (268, 354) and <268; in 2002, the first to fourth classes were classified as USD ≥1569, (958, 1569), (743, 958) and <743; and in 2012, the first to fourth classes were classified as USD ≥8510, (5761, 8510), (4670, 5761) and <4670.

The series of national surveys was approved by the ethics committee of the National Institute for Nutrition and Health at the Chinese Center for Disease Control and Prevention.

### 2.2. Dietary Assessment

The field work of each round was launched in autumn, considering the comparability between survey rounds. Dietary information was collected for 5 days in 1982 by trained investigators who weighed all available foods in the participants’ homes at the beginning of the first day, recorded (and weighed if necessary) all new foods brought into the homes during the 5 days and weighed all leftovers at the end of the survey to calculate the total amount of food consumed by participants during those 5 days. In the 1992, 2002 and 2010–2012 surveys, diet was assessed via 3 consecutive days (including two weekdays and one weekend) of 24-h dietary recall, in addition to weighing household cooking oil and condiments. For each dietary recall day, investigators went to participants’ homes and helped to record food intake during the past 24 h. Investigators also weighed the household cooking oil and condiments at the beginning and end of each 24 h dietary survey. Nutrient intakes were calculated with the China Food Composition tables (FCTs) [6,7,8], which are continuously updated with commonly consumed foods and changes in nutrient composition. FCT-1981 [8] was used for dietary data from the 1982 round, FCT-2002 [6] for those from the 1992 and 2002 rounds, and FCT-2009 [7] for those from the 2010–2012 round. 

### 2.3. Statistical Analyses

The post-stratification population sampling weights were applied to the estimated nationally representative population levels for intakes of energy and macronutrients. In order to compare dietary intake across years, the weights were derived from the sampling probability of the 2010 Chinese population aged 20 years or older (based on census data) and applied to estimate the representative dietary intake in each survey round. Means and 95% confidence intervals (CIs) of energy, and the percentages of macronutrients contributing to energy, were determined by adjustment for the sample weights. General linear regression models were used to determine the dietary trends across the survey rounds and the dietary differences between and within years. Regarding the difference between years, the year of each survey was treated as an ordinal variable and as the dependent variable. Regarding the difference within years, the subgroup of the two ends within each group (classified by education level, area, GNP level, sex and age group) was treated as an ordinal variable and as the dependent variable. Energy and macronutrient composition were treated as continuous variables and as the independent variable respectively in each model. A two-sided *p* < 0.05 was considered to indicate statistical significance. Statistical analyses were conducted using SAS statistical software (v. 9.4; SAS Institute, Cary, NC, USA).

## 3. Results

### 3.1. Participant Characteristics

There were 204,877 participants aged 20 years or older included in the current analysis. In the survey rounds of 1982, 1992, 2002 and 2010–2012, dietary intake data were available for 39,084, 58,316, 52,426 and 55,051 participants, respectively. The age structure of the participants was assorted across the survey rounds in accordance with the structure of actual change among the Chinese population. The sex ratios were balanced in the samples. Participants in each round had higher education level than the former round. Urban participants gradually accounted for greater percentages of the samples in the survey rounds, due to urbanization progress in China (Table 1).

### 3.2. Trends of Energy and Macronutrient Composition 

From 1982 to 2012, the estimated energy intake declined from 2614.7 kcal to 2063.9 kcal (1992 vs. 1982 difference, −82.6; 95% CI −92.5 to −72.7; 2002 vs. 1992 difference, −335.4; 95% CI −344.2 to −326.7; 2012 vs. 2002 difference, −132.7; 95% CI −141.5 to −123.9; *p* < 0.01 for trend). These were the trends in the population subgroups (Table 2).

The trend of estimated percentage of energy intake from fat showed a spike. It increased from 16.3% to 33.1% (1992 vs. 1982 difference, 7.6%; 95% CI 7.4% to 7.7%; 2002 vs. 1992 difference, 7.7%; 95% CI 7.6% to 7.9%; 2012 vs. 2002 difference, 1.6%; 95% CI 1.4% to 1.7%; *p* < 0.01 for trend). The trends coincided in the subgroups (all *p* < 0.01 for trend) except for the subgroup of those educated for over 15 years. In the most recent two survey rounds, the estimated percentage of energy intake from fat among the well-educated population declined from 37.4% to 36.6% (2012 vs. 2002 difference, −0.8%; 95% CI −1.6% to 0.0%) (Table 3).

The estimated percentage of energy intake from carbohydrates declined from 74.0% to 55.0% (1992 vs. 1982 difference, −10.5%; 95% CI −10.7% to −10.4%; 2002 vs. 1992 difference, −7.4%; 95% CI −7.5% to −7.2%; 2012 vs. 2002 difference, −1.0%; 95% CI −1.2% to −0.9%; *p* < 0.01 for trend). The trends in the subgroups were the same (Table 4).

The estimated percentage of energy intake from protein increased between the first and second rounds, from 10.9% to 12.8%, and slightly declined to 12.3% in the successive two rounds (1992 vs. 1982 difference, 1.9%; 95% CI 1.9% to 1.9%; 2002 vs. 1992 difference, −0.3%; 95% CI −0.4% to −0.3%; 2012 vs. 2002 difference, −0.1%; 95% CI −0.1% to 0.0%) (Table 5).

### 3.3. Disparities of Macronutrient Composition in Population Subgroups

The estimated percentages of energy intake from fat within population subgroups, stratified by education level, were 10.6% (95% CI 10.1–11.1%) in 1992, 9.5% (95% CI 8.8–10.1%) in 2002 and 6.3% (95% CI 5.8–6.7%) in 2010–2012. Those stratified by area were 6.8% (95% CI 6.7–7.0%) in 1982, 10.8% (95% CI 10.6–10.9%) in 1992, 8.9% (95% CI 8.7–9.1%) in 2002 and 7.4% (95% CI 7.2–7.6%) in 2010–2012. Those stratified by GNP level were 2.0% (95% CI 1.8–2.3%) in 1982, 7.3% (95% CI 7.1–7.6%) in 1992, 2.9% (95% CI 2.6–3.1%) in 2002 and 3.7% (95% CI 3.5–4.0%) in 2010–2012. The ranges of the estimated percentage of energy intake from carbohydrates within population subgroups stratified by education level were 13.1% (95% CI 12.6–13.6%) in 1992, 12.4% (95% CI 11.8–13.0%) in 2002 and 9.5% (95% CI 9.0–10.0%) in 2010–2012. Those stratified by area were 8.2% (95% CI 8.0–8.3%) in 1982, 12.9% (95% CI 12.7–13.0%) in 1992, 10.8% (95% CI 10.7–11.0%) in 2002 and 9.6% (95% CI 9.4–9.8%) in 2010–2012. Those stratified by GNP level were 2.9% (95% CI 2.6–3.2%) in 1982, 9.2% (95% CI 9.0–9.5%) in 1992, 4.8% (95% CI 4.5–5.1%) in 2002 and 5.8% (95% CI 5.5–6.1%) in 2010–2012 (Table 2, Table 3, Table 4 and Table 5 and Figure 1). The trends and disparities stratified by age and sex are given in Table A1, Table A2, Table A3 and Table A4.

## 4. Discussion

China has made substantial progress in improving nutrition. Diet quality improved remarkably from 1982 to 2012 in China. The trends of energy intake constantly decreased in the survey rounds due to the fast pace of modernization and urbanization. The percentage of fat’s contribution to energy spiked, that of carbohydrates fell all the way, and that of protein stabilized within a small range. The macronutrient composition went from poor, to ideal, and then to far from ideal again. Though the composition was not satisfying at the beginning round of CNNS, in 1982, which featured excessive carbohydrates and a lack of fat, it became more ideal in the 1992 survey round. The macronutrient composition was within the national recommendations among most subpopulations around that period [9]. However, in the most recent two surveys, the macronutrient composition dropped out of the ideal range, which led to health conditions diametrically opposed to malnutrition, i.e., overnutrition, potentially contributing to the prevalence of nutrition-related non-communicable chronic diseases (NCDs) nation-wide [10]. We considered that different fat compositions at the same level of energy intake could have diverse impacts on the development of obesity. It seemed a paradox in China that overweight and obesity dramatically increased since 1980s, despite energy intake constantly decreasing [1,11]. Reduced physical activity could explain the increasing prevalence of obesity, but most developed countries, like America and Korea, also experience both obesity prevalence and raising energy intakes [12,13,14,15]. Indeed, few countries, like Japan, had a similar situation to ours, whereby the obesity rate went up as the energy intake decreased [16]. New studies suggested that the percentage of fat contributing to energy could be the cause of adiposity, but not carbohydrates or protein [17]. In fact, the proportion of fat in the diet kept going up worldwide, as did the prevalence of obesity [13,16,18,19]. The current findings were based on massive samples and observations over the long-term, which might provide new thoughts as to the cause of obesity. 

The great achievements following the social and economic shifts after 1979 had a tremendous impact on the diet of the Chinese population [20]. It took no more than one decade for the Chinese people to go from lacking various foods, to having plenty of every food. There was a big leap in nutrition improvement, and diet patterns changed most in the 1980s and 1990s. The macronutrient composition rapidly reached the ideal range at that time. The pace of the change of macronutrient composition slowed down, and it has been unsatisfying in recent years. The promoter of the diet has shifted. Economy and food supply were still continuing to improve, but it was contributing only a little to diet improvement in China. Other things, like nutrition policy retargeting or the availability of nutrition education and knowledge, might be the key to promoting diet quality in China. 

The disparities persistently existed in different subpopulations across China, but the gaps narrowed in recent years. The Chinese government has put huge effort into poverty reduction, transportation system construction and raising the agricultural yield, which all potentially increased the equity of access to various foods by people with different background. Especially in the most recent survey round, the percentages of fats’ and carbohydrates’ contributions to energy were getting closer between the two ends of the subpopulations as regards area, education level and economic background. It was obvious that the subpopulations with better social profiles (living in urban areas, well-educated and wealthy economic background) were leading the diet trends, and the rest followed in the next decade or two. Nevertheless, the macronutrient compositions of those with better social profiles had been moving toward the overnutrition pattern since around 2002, which was probably a major cause of nutrition-related NCDs prevailing in China [3,10,21]. If people with low social profiles continue to follow the diet trend, there might be another surge of nutrition-related NCDs in China. Moreover, inequalities in health resource access have existed for some time in China [22]. It would deepen the social contradictions if those who suffered from diseases could not be able to access necessary health resources. More governmental interventions should be launched into the subpopulation with low social profiles in order to slow down or even curb their movement into overnutrition. 

One promising trend was discovered in the well-educated subpopulation. In the survey round of 2012, the macronutrient composition distinctively retuned to the recommended ranges among these people. “Eat well” was linked to “live well” in Chinese culture, but people always confused “eat well” with “eat whatever one wants”. Actually, “eat well” means “eat properly” in the modern nutritional theory, and it leads to “live well”. It is clear that some risks of nutrition-related NCDs can be modified though education improvement [23]. Well-educated people have greater volition and ability to acquire health information which might help them regulate dietary behaviors, rather than following their instinctive appetite or preference. Health education would probably be a useful tool to help China get through the possible dilemma of a further potential surge of NCDs in the subpopulations with low social profiles.

This study has several limitations. First, 3-day 24-h dietary recalls were used to obtain food consumption information, and so the accuracy of dietary intake was mostly dependent on the participants’ recall and estimation. Second, for the individual income information variabilities in different survey rounds, the classification of GNP level was applied. It was based on each province’s GNP in the survey year, which might not classify each participant meticulously. Third, a recent study mentioned that the quality or food sources of macronutrients might lead to different health outcomes [24]. Diet quality in the current study was determined based on the macronutrient composition, which might cause bias without taking food composition into consideration. Fourth, the inference of macronutrient composition and consequential health outcome in the discussion was only derived from reports on the national level in an ecological way, rather than the relationships among CNNS participants.

## 5. Conclusions

Quick improvements in society and the economy effectively curbed undernutrition, but easily triggered larger-scaled overnutrition soon after in China. Disparities have persistently existed in different subpopulations, while these gaps would narrow if major efforts were made. Populations with low social profiles might lag behind the trend of diet transition, but would be more vulnerable to the side-effects of the trend. Education might be a promising way of preventing overnutrition during the prosperous progress of developing countries. Low social profile populations require specific interventions so as to avoid the further burdens of diet-related non-communicable diseases, in order to maintain social stability.

## Figures and Tables

**Figure 1 nutrients-12-02168-f001:**
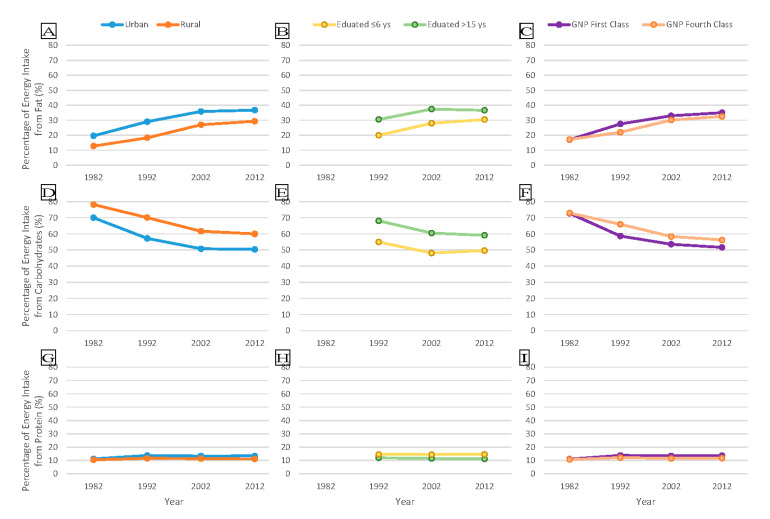
Trends and Disparities between Two Ends of Subgroups with regards to Estimated Energy Intake from Macronutrients of Adults Aged 18 Years or Older by CNNS Round, 1982–2012, Stratified by Area, Education Level and GNP Level. The two polygonal lines in each graph represent the two ends of subgroups. (**A**–**C**) show percentages of energy intake from fat across survey rounds. (**D**–**F**) show percentages of energy intake from carbohydrates across survey rounds. (**G**–**I**) show percentages of energy intake from protein across survey rounds.

**Table 1 nutrients-12-02168-t001:** Sociodemographic Characteristics of Participants by China National Nutrition Survey (CNNS) Rounds, 1982–2012.

		1982	1992	2002	2012
*n*		39,084	58,316	52,426	55,051
Age Group, year					
	20–29	12,642 (32.4)	16,116 (27.6)	7531 (14.4)	5310 (9.7)
	30–39	8729 (22.3)	13,840 (23.7)	12,959 (24.7)	7894 (14.3)
	40–49	6540 (16.7)	11,440 (19.6)	11,745 (22.4)	12,420 (22.6)
	50–59	5533 (14.2)	8429 (14.5)	10,201 (19.5)	12,828 (23.3)
	60–69	3662 (9.4)	5598 (9.6)	6630 (12.7)	10,308 (18.7)
	≥70	1978 (5.1)	2893 (5.0)	3360 (6.4)	6291 (11.4)
Sex					
	Male	19,432 (49.7)	28,010 (48.0)	24,709 (47.1)	25,278 (45.9)
	Female	19,652 (50.3)	30,306 (52.0)	27,717 (52.9)	29,773 (54.1)
Education Level					
	under 6 years		23,479 (40.3)	6567 (12.5)	6901 (12.5)
	6 years		8477 (14.5)	15,686 (29.9)	15,866 (28.8)
	9 years		15,620 (26.8)	18,075 (34.5)	19,064 (34.6)
	12 years		7766 (13.3)	8433 (16.1)	8454 (15.4)
	15 years		1514 (2.6)	2452 (4.7)	2740 (5.0)
	over 15 years		1148 (2.0)	1115 (2.1)	2026 (3.7)
	No answer		312 (0.5)	98 (0.2)	0 (0.0)
Area					
	Urban	13,766 (35.2)	17,633 (30.2)	17,530 (33.4)	27,471 (49.9)
	Rural	25,318 (64.8)	40,683 (69.8)	34,896 (66.6)	27,580 (50.1)
GNP Level ^1^					
	First Class	7195 (18.4)	16,635 (28.5)	15,374 (29.3)	15,384 (27.9)
	Second Class	11,758 (30.1)	13,802 (23.7)	10,916 (20.8)	14,244 (25.9)
	Third Class	8508 (21.8)	14,544 (24.9)	14,693 (28.0)	13,818 (25.1)
	Fourth Class	11,623 (29.7)	13,335 (22.9)	11,443 (21.8)	11,605 (21.1)

CNNS, China National Nutrition Survey; GNP, gross national product. Data are numbers of participants (%), unless otherwise indicated. ^1^ GNP level was classified at provincial level according to the GNP quartiles across provinces. In 1982, the first to fourth classes were classified as USD ≥284, (244, 284), (194, 244) and <194, respectively; in 1992, the first to fourth class were classified as USD ≥482, (354, 482), (268, 354) and <268; in 2002, the first to fourth classes were classified as USD ≥1569, (958, 1569), (743, 958) and <743; and in 2012, the first to fourth classes were classified as USD ≥8510, (5761, 8510), (4670, 5761) and <4670.

**Table 2 nutrients-12-02168-t002:** Trends and Disparities in the Daily Energy Intake of Adults Aged 18 Years or Older by CNNS Round, 1982–2012 ^1^.

		Daily Energy Intake-Survey-Weighted Mean, kcal (95% CI)				*p* Value for Trend	Difference between Rounds, kcal (95% CI)		
		1982	1992	2002	2012		1992 vs. 1982	2002 vs. 1992	2012 vs. 2002
All		2614.7 (2606.5-2622.8)	2532.1 (2525.9–2538.2)	2196.6 (2190.4–2202.8)	2063.9 (2057.7–2070.2)	<0.01	−82.6 (−92.5 to −72.7)	−335.4 (−344.2 to −326.7)	−132.7 (−141.5 to −123.9)
Education level									
	under 6 years		2488.7 (2478.3–2499.1)	2055.7 (2038.5–2072.8)	1882.0 (1865.7–1898.4)	<0.01		−433.0 (−454.2 to −411.8)	−173.6 (−197.4 to −149.9)
	6 years		2618.8 (2602.5–2635.1)	2288.5 (2277.2–2299.8)	2126.4 (2114.4–2138.3)	<0.01		−330.3 (−349.9 to −310.8)	−162.1 (−178.5 to −145.6)
	9 years		2589.0 (2577.4–2600.6)	2269.2 (2258.6–2279.9)	2155.5 (2144.5–2166.5)	<0.01		−319.8 (−335.6 to −303.9)	−113.8 (−129.1 to −98.4)
	12 years		2494.3 (2479.0–2509.6)	2123.3 (2108.2–2138.4)	2015.2 (2000.3–2030.2)	<0.01		−371.0 (−392.6 to −349.5)	−108.1 (−129.4 to −86.7)
	15 years		2500.7 (2467.0–2534.4)	2052.3 (2025.4–2079.2)	1911.1 (1886.9–1935.3)	<0.01		−448.4 (−491.6 to −405.1)	−141.2 (−177.3 to −105.2)
	over 15 years		2472.2 (2436.3–2508.0)	2035.2 (1996.9–2073.5)	1883.3 (1855.6–1911.1)	<0.01		−437.0 (−489.5 to −384.5)	−151.9 (−198.6 to −105.2)
	Range within subgroups		146.6 (109.8–183.4)	253.3 (220.9–285.6)	273.4 (250.7–296.2)				
Area									
	Urban	2531.0 (2518.1–2543.9)	2423.6 (2413.5–2433.7)	1983.9 (1974.2–1993.7)	1897.2 (1889.4–1905.0)	<0.01	−107.4 (−123.5 to −91.2)	−439.7 (−453.8 to −425.7)	−86.7 (−98.9 to −74.6)
	Rural	2703.9 (2693.3–2714.5)	2647.7 (2639.8–2655.6)	2423.5 (2416.0–2431.0)	2241.8 (2232.5–2251.1)	<0.01	−56.2 (−68.9 to −43.4)	−224.2 (−235.1 to −213.3)	−181.7 (−193.5 to −169.9)
	Range within subgroups	172.9 (156.7–189.1)	224.1 (211.9–236.2)	439.6 (427.8–451.4)	344.6 (332.5–356.7)				
GNP level									
	First class	2624.2 (2605.9–2642.5)	2443.4 (2432.9–2453.8)	2173.6 (2162.9–2184.3)	1963.9 (1953.2–1974.6)	<0.01	−180.8 (−200.0 to −161.6)	−269.7 (−284.7 to −254.8)	−209.7 (−224.8 to −194.5)
	Second class	2555.3 (2540.6–2570.0)	2514.8 (2502.3–2527.4)	2140.9 (2127.7–2154.2)	1936.6 (1925.8–1947.5)	<0.01	−40.5 (−59.8 to −21.1)	−373.9 (−392.1 to −355.7)	−204.3 (−221.2 to −187.4)
	Third class	2629.1 (2611.2–2647.1)	2538.4 (2526.3–2550.5)	2191.5 (2179.7–2203.4)	2141.6 (2128.8–2154.4)	<0.01	−90.7 (−111.3 to −70.1)	−346.9 (−363.9 to −329.9)	−49.9 (−67.3 to −32.5)
	Fourth class	2654.6 (2639.7–2669.6)	2681.6 (2667.2–2696.0)	2299.3 (2285.0–2313.6)	2260.3 (2245.1–2275.6)	<0.01	27.0 (6.1 to 48.0)	−382.3 (−402.7 to −361.9)	−39.0 (−59.9 to −18.1)
	Range within subgroups	99.3 (78.4–120.3)	238.3 (220.7–255.8)	158.4 (138.9–177.9)	323.7 (305.5–342.0)				

CNNS, China National Nutrition Survey; GNP, gross national product. ^1^ Data were adjusted for CNNS weights to be nationally representative. Values may not equal the difference between two years’, or the highest and lowest subgroups’, estimates because of rounding.

**Table 3 nutrients-12-02168-t003:** Trends and Disparities in the Estimated Percentage of Energy Intake from Fat of Adults Aged 18 Years or Older by CNNS Round, 1982–2012 ^1^.

		Estimated Percentage of Energy Intake from Fat, Survey–Weighted % (95% CI)				*p* Value for Trend	Difference between Rounds, % (95% CI)		
		1982	1992	2002	2012		1992 vs. 1982	2002 vs. 1992	2012 vs. 2002
All		16.3 (16.2–16.4)	23.8 (23.7–23.9)	31.6 (31.5–31.7)	33.1 (33.0–33.2)	<0.01	7.6 (7.4 to 7.7)	7.7 (7.6 to 7.9)	1.6 (1.4 to 1.7)
Education level									
	under 6 years		19.9 (19.8–20.0)	28.0 (27.7–28.3)	30.4 (30.1–30.7)	<0.01		8.1 (7.8 to 8.4)	2.5 (2.0 to 2.9)
	6 years		22.2 (22.0–22.5)	28.9 (28.7–29.1)	31.1 (31.0–31.3)	<0.01		6.6 (6.3 to 6.9)	2.3 (2.0 to 2.5)
	9 years		25.2 (25.0–25.3)	31.0 (30.8–31.2)	32.9 (32.7–33.1)	<0.01		5.8 (5.6 to 6.1)	1.9 (1.7 to 2.1)
	12 years		28.2 (28.0–28.4)	34.8 (34.6–35.1)	35.1 (34.8–35.3)	<0.01		6.6 (6.2 to 6.9)	0.3 (−0.1 to 0.6)
	15 years		30.3 (29.8–30.8)	36.5 (36.1–37.0)	36.7 (36.3–37.1)	<0.01		6.2 (5.5 to 6.9)	0.1 (−0.5 to 0.7)
	over 15 years		30.5 (30.0–31.0)	37.4 (36.8–38.0)	36.6 (36.2–37.1)	<0.01		6.9 (6.1 to 7.7)	−0.8 (−1.6 to 0.0)
	Range within subgroups		10.6 (10.1–11.1)	9.5 (8.8–10.1)	6.3 (5.8–6.7)				
Area									
	Urban	19.6 (19.4–19.7)	29.0 (28.9–29.2)	35.8 (35.7–36.0)	36.7 (36.6–36.8)	<0.01	9.5 (9.3 to 9.7)	6.8 (6.6 to 7.0)	0.9 (0.7 to 1.1)
	Rural	12.7 (12.6–12.8)	18.3 (18.2–18.4)	27.0 (26.9–27.1)	29.3 (29.2–29.4)	<0.01	5.5 (5.4 to 5.7)	8.7 (8.6 to 8.9)	2.3 (2.1 to 2.5)
	Range within subgroups	6.8 (6.7–7.0)	10.8 (10.6–10.9)	8.9 (8.7–9.1)	7.4 (7.2–7.6)				
GNP level									
	First class	17.0 (16.8–17.2)	27.5 (27.4–27.7)	33.0 (32.8–33.1)	35.0 (34.9–35.2)	<0.01	10.5 (10.3 to 10.8)	5.4 (5.2 to 5.7)	2.1 (1.8 to 2.3)
	Second class	15.8 (15.6–15.9)	23.7 (23.5–23.9)	31.2 (31.0–31.4)	33.3 (33.1–33.5)	<0.01	8.0 (7.7 to 8.2)	7.5 (7.2 to 7.7)	2.1 (1.8 to 2.4)
	Third class	15.1 (14.9–15.3)	20.2 (20.0–20.4)	31.3 (31.1–31.5)	31.3 (31.1–31.5)	<0.01	5.1 (4.8 to 5.3)	11.1 (10.8 to 11.4)	0.0 (−0.3 to 0.3)
	Fourth class	17.1 (17.0–17.3)	21.9 (21.7–22.1)	30.1 (29.9–30.3)	32.5 (32.3–32.7)	<0.01	4.8 (4.5 to 5.1)	8.2 (7.9 to 8.5)	2.4 (2.1 to 2.7)
	Range within subgroups	2.0 (1.8–2.3)	7.3 (7.1–7.6)	2.9 (2.6–3.1)	3.7 (3.5–4.0)				

CNNS, China National Nutrition Survey; GNP, gross national product. ^1^ Data were adjusted for CNNS weights to be nationally representative. Values may not equal the difference between two years’, or the highest and lowest subgroups’, estimates because of rounding.

**Table 4 nutrients-12-02168-t004:** Trends and Disparities in the Estimated Percentage of Energy Intake from Carbohydrates of Adults Aged 18 Years or Older by CNNS Round, 1982–2012 ^1^.

		Estimated Percentage of Energy Intake from Carbohydrates, Survey–Weighted % (95% CI)				*p* Value for Trend	Difference between Rounds, % (95% CI)		
		1982	1992	2002	2012		1992 vs. 1982	2002 vs. 1992	2012 vs. 2002
All		74.0 (73.8–74.1)	63.4 (63.3–63.5)	56.0 (55.9–56.1)	55.0 (54.9–55.1)	<0.01	−10.5 (−10.7 to −10.4)	−7.4 (−7.5 to −7.2)	−1.0 (−1.2 to −0.9)
Education level									
	under 6 years		68.1 (67.9–68.2)	60.5 (60.2–60.8)	59.1 (58.8–59.4)	<0.01		−7.6 (−7.9 to −7.3)	−1.4 (−1.8 to −1.0)
	6 years		65.4 (65.2–65.6)	59.5 (59.3–59.7)	57.7 (57.5–57.9)	<0.01		−5.9 (−6.2 to −5.6)	−1.8 (−2.0 to −1.5)
	9 years		61.9 (61.8–62.1)	56.8 (56.6–57.0)	55.4 (55.2–55.6)	<0.01		−5.1 (−5.4 to −4.9)	−1.4 (−1.7 to −1.2)
	12 years		58.1 (57.9–58.4)	51.9 (51.7–52.2)	52.4 (52.1–52.6)	<0.01		−6.2 (−6.5 to −5.8)	0.4 (0.1 to 0.8)
	15 years		55.4 (54.9–55.9)	49.6 (49.2–50.0)	50.0 (49.6–50.4)	<0.01		−5.8 (−6.5 to −5.1)	0.4 (−0.2 to 1.0)
	over 15 years		55.0 (54.4–55.5)	48.1 (47.5–48.7)	49.6 (49.2–50.1)	<0.01		−6.9 (−7.7 to −6.1)	1.5 (0.8 to 2.3)
	Range within subgroups		13.1 (12.6–13.6)	12.4 (11.8–13.0)	9.5 (9.0–10.0)				
Area									
	Urban	70.0 (69.8–70.2)	57.2 (57.0–57.3)	50.8 (50.6–51.0)	50.4 (50.2–50.5)	<0.01	−12.8 (−13.0 to −12.6)	−6.4 (−6.6 to −6.2)	−0.4 (−0.6 to −0.2)
	Rural	78.2 (78.0–78.3)	70.1 (70.0–70.2)	61.6 (61.5–61.7)	60.0 (59.8–60.1)	<0.01	−8.1 (−8.2 to −7.9)	−8.4 (−8.6 to −8.3)	−1.7 (−1.8 to −1.5)
	Range within subgroups	8.2 (8.0–8.3)	12.9 (12.7–13.0)	10.8 (10.7–11.0)	9.6 (9.4–9.8)				
GNP level									
	First class	72.7 (72.5–73.0)	58.7 (58.5–58.8)	53.6 (53.4–53.7)	51.6 (51.4–51.8)	<0.01	−14.1 (−14.3 to −13.8)	−5.1 (−5.3 to −4.9)	−1.9 (−2.2 to −1.7)
	Second class	74.5 (74.3–74.7)	63.7 (63.5–63.9)	56.5 (56.3–56.8)	55.4 (55.2–55.6)	<0.01	−10.8 (−11.0 to −10.5)	−7.2 (−7.5 to −6.9)	−1.1 (−1.4 to −0.8)
	Third class	75.6 (75.4–75.8)	67.9 (67.7–68.0)	56.8 (56.6–57.0)	57.4 (57.2–57.6)	<0.01	−7.8 (−8.0 to −7.5)	−11.1 (−11.3 to −10.8)	0.6 (0.3 to 0.9)
	Fourth class	72.9 (72.7–73.1)	65.9 (65.7–66.1)	58.4 (58.1–58.6)	56.3 (56.0–56.5)	<0.01	−7.0 (−7.3 to −6.7)	−7.6 (−7.9 to −7.2)	−2.1 (−2.4 to −1.8)
	Range within subgroups	2.9 (2.6–3.2)	9.2 (9.0–9.5)	4.8 (4.5–5.1)	5.8 (5.5–6.1)				

CNNS, China National Nutrition Survey; GNP, gross national product. ^1^ Data were adjusted for CNNS weights to be nationally representative. Values may not equal the difference between two years’, or the highest and lowest subgroups’, estimates because of rounding.

**Table 5 nutrients-12-02168-t005:** Trends and Disparities in the Estimated Percentage of Energy Intake from Protein of Adults Aged 18 Years or Older by CNNS Round, 1982–2012 ^1^.

		Estimated Percentage of Energy Intake from Protein, Survey-Weighted % (95% CI)				*p* Value for Trend	Difference between Rounds, % (95% CI)		
		1982	1992	2002	2012		1992 vs. 1982	2002 vs. 1992	2012 vs. 2002
All		10.9 (10.8–10.9)	12.8 (12.7–12.8)	12.4 (12.4–12.4)	12.3 (12.3–12.4)	<0.01	1.9 (1.9 to 1.9)	−0.3 (−0.4 to −0.3)	−0.1 (−0.1 to 0.0)
Education level									
	under 6 years		12.0 (12.0–12.0)	11.5 (11.5–11.6)	11.2 (11.2–11.3)	<0.01		−0.5 (−0.6 to −0.4)	−0.3 (−0.4 to −0.2)
	6 years		12.4 (12.3–12.4)	11.7 (11.6–11.7)	11.5 (11.4–11.5)	<0.01		−0.7 (−0.8 to −0.6)	−0.2 (−0.2 to −0.1)
	9 years		12.9 (12.9–13.0)	12.2 (12.2–12.2)	12.1 (12.0–12.1)	<0.01		−0.7 (−0.8 to −0.6)	−0.1 (−0.2 to −0.1)
	12 years		13.7 (13.6–13.7)	13.2 (13.2–13.3)	13.2 (13.1–13.2)	<0.01		−0.4 (−0.5 to −0.3)	−0.1 (−0.2 to 0.0)
	15 years		14.2 (14.0–14.4)	13.8 (13.7–14.0)	14.1 (14.0–14.3)	0.83		−0.4 (−0.6 to −0.2)	0.3 (0.1 to 0.5)
	over 15 years		14.5 (14.3–14.7)	14.5 (14.3–14.7)	14.6 (14.4–14.8)	0.40		0.0 (−0.3 to 0.2)	0.1 (−0.1 to 0.4)
	Range within subgroups		2.5 (2.4–2.6)	3.0 (2.8–3.1)	3.4 (3.3–3.5)				
Area									
	Urban	11.2 (11.2–11.3)	13.8 (13.7–13.8)	13.4 (13.3–13.4)	13.5 (13.4–13.5)	<0.01	2.6 (2.5 to 2.6)	−0.4 (−0.5 to −0.3)	0.1 (0.0 to 0.2)
	Rural	10.5 (10.5–10.5)	11.7 (11.6–11.7)	11.4 (11.4–11.4)	11.2 (11.1–11.2)	<0.01	1.2 (1.1 to 1.2)	−0.3 (−0.3 to −0.2)	−0.2 (−0.3 to −0.2)
	Range within subgroups	0.7 (0.7–0.8)	2.1 (2.1–2.1)	2.0 (1.9–2.0)	2.3 (2.3–2.4)				
GNP level									
	First class	11.0 (10.9–11.0)	13.8 (13.8–13.9)	13.5 (13.4–13.5)	13.6 (13.5–13.7)	<0.01	2.8 (2.8 to 2.9)	−0.3 (−0.4 to −0.3)	0.1 (0.0 to 0.2)
	Second class	11.2 (11.2–11.2)	12.5 (12.5–12.6)	12.3 (12.2–12.3)	12.0 (12.0–12.1)	<0.01	1.3 (1.3 to 1.4)	−0.3 (−0.3 to −0.2)	−0.3 (−0.3 to −0.2)
	Third class	10.6 (10.5–10.6)	12.0 (11.9–12.0)	11.9 (11.9–12.0)	11.8 (11.8–11.9)	<0.01	1.4 (1.3 to 1.4)	0.0 (−0.1 to 0.0)	−0.1 (−0.2 to 0.0)
	Fourth class	10.7 (10.7–10.7)	12.1 (12.1–12.2)	11.5 (11.5–11.6)	11.7 (11.6–11.8)	<0.01	1.4 (1.4 to 1.5)	−0.6 (−0.7 to −0.5)	0.2 (0.1 to 0.2)
	Range within subgroups	0.6 (0.5–0.7)	1.9 (1.8–1.9)	1.9 (1.9–2.0)	1.9 (1.8–2.0)				

CNNS, China National Nutrition Survey; GNP, gross national product. ^1^ Data were adjusted for CNNS weights to be nationally representative. Values may not equal the difference between two years’, or the highest and lowest subgroups’, estimates because of rounding.

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
