# Peer review of "Trends and Disparities of Energy Intake and Macronutrient Composition in China: A Series of National Surveys, 1982–2012"

_nutrients, 2020, doi:10.3390/nu12082168_

Round 1

Reviewer 1 Report

Zhu et al surveyed the Chinese population on energy intake and identified disparities persistently existed in different subpopulations. Study revealed that quick improvements in society and economy effectively curbed undernutrition but easily triggered overnutrition. Specific interventions and Education might be a promising way to preventing overnutrition during the prosperous progress.

Author need to expand the conclusion section that can better reflects the result.   

Overall, this study is very meaningful and can be used as a model for developing countries across the world. 

Author Response

Thank you for the pertinent comments. Reply: The Conclusion Part was revised as bellow: Quick improvements in society and economy effectively curbed undernutrition but easily triggered larger-scaled overnutrition soon after in China. Disparities persistently existed in different subpopulations while the gaps would narrow if major efforts put on. Population with low social profile might lag the trend of diet transition but would be more vulnerable to the side-effect of the trend. Education might be a promising way to preventing overnutrition during the prosperous progress in developing countries. Low social profile population required specific interventions to avoid further burden of diet-related non-communicable disease in order to maintain social stability.

Reviewer 2 Report

Overall the paper requires fairly straightforward statistical methods.
A little more detail is needed in the statement "General linear regression models
were used to determine the dietary trends across the survey rounds and the dietary
differences between and within years.". The dependent variable should be specified,
and the predictor variables listed, including whether they were continuous, ordinal
or categorical. This is stated for year, but not for whatever other variables were
included to look at differences within year.

It needs to be clearer how the comparison across years accounts for change in
the demographics of the sample. We see in Table 1 for example that the age
distribution is gradually getting older. This may affect diet if this changes
with age. Sample weights will not deal with this as the population is likely
to be changing, not just the sample.

Author Response

Overall the paper requires fairly straightforward statistical methods.
A little more detail is needed in the statement "General linear regression models
were used to determine the dietary trends across the survey rounds and the dietary
differences between and within years.". The dependent variable should be specified,
and the predictor variables listed, including whether they were continuous, ordinal
or categorical. This is stated for year, but not for whatever other variables were
included to look at differences within year.

 Reply:

A more detailed description was given in the Methods Part “2.3 Statistical Analyses” as below:

Regarding the difference between years, the year of each survey was treated as an ordinal variable and being the dependent variable. Regarding the difference within years, the subgroup of the two ends within each group (classified by education level, area, GNP level, sex and age group) was treated as an ordinal variable and being the dependent variable. Energy and macronutrient composition were treated as continuous variables and being the independent variable respectively in each model.

It needs to be clearer how the comparison across years accounts for change in
the demographics of the sample. We see in Table 1 for example that the age
distribution is gradually getting older. This may affect diet if this changes
with age. Sample weights will not deal with this as the population is likely
to be changing, not just the sample.

 Reply:

Though the age distribution was changing across the survey rounds, we had adjusted the dietary intake representing the intake of a fixed population. A detailed explanation was given in the Methods Part “2.3 Statistical Analyses”:

The post-stratification population sampling weights were applied to estimated nationally representative population levels for intakes of energy and macronutrient. In order to compare the dietary intake across years, the weights were derived from the sampling probability of the 2010 Chinese population aged 20 years or older (based on census data) and applied to estimate the representative dietary intake in each survey round.

Reviewer 3 Report

REVIEW: Trends and Disparities of Energy Intake and Macronutrient Composition in China: a series of national surveys, 1982-2012

Thank you for providing me an opportunity to review a paper submitted to Nutrients.

In this paper, the authors described 30-years (every 10-year) time trend of nutritional status in the Chinese population, showed disparities of energy intake according to social factors, and provided several suggestions for further issues of diet-related non-communicable diseases. The manuscript is well-written without redundant description.

I have some minor comments/questions.

  1. Abstract section:

Suggest change from CNNS (abbreviation) to “China National Nutrition Survey”. Readers outside China do not know CNNS and confuse when first looking at this abbreviation in the abstraction section.

  1. Methods section:

I could not clearly understand the participant selection of the study. There were initially 238 124, 100 201, 247 464, and 188 622 participants recruited in the series of surveys from 1982 to 2012, respectively. Data were available for 39 084, 58 316, 52 426, and 55 051 participants, respectively. In each year, 16% (39084/238124), 58% (58316/100201), 21% (52426/247464), and 29% (55051/188622) of participants were included in the analysis. Is this correct? Why the percentages of participant selection largely differ? I feel most of the participants of the surveys excluded from the analysis. Why? I suggest showing participant selection using a brief figure.

  1. Methods section:

What was the definition of urban or rural areas? Is this just based on the population?

  1. Results section (Table 1):

It will become clearer to know population aging and change in education levels if the authors add means (or medians) of the variables in Table 1.

  1. Results section (Table 2):

Table 2 is not totally shown. Table 2 should be simple to delete 1992 vs …. (all columns to the right of “p for trend”) and “Range” and “subgroups” lines. I can see these results described in the main text.

  1. Results section (Table 3-5):

Same as comments for Table 2 above, the columns “Difference between Rounds, % (95%CI)” seems to be unnecessary. These results may describe in the main text only, if necessary. I think it is not unnecessary showing “Range”, lines as well.

  1. Results section (description for Table 3):

Lines 126-127: ~except the subgroup of those educated over 15 years.

I could not understand why except those educated over 15 years because of trend p <0.01.

  1. Discussion section:

Line 20:

“NCD” should be expanded, followed by the abbreviation. This is the first use in the manuscript.

Author Response

Thank you for the pertinent comments.

  1. Abstract section:

Suggest change from CNNS (abbreviation) to “China National Nutrition Survey”. Readers outside China do not know CNNS and confuse when first looking at this abbreviation in the abstraction section.

 Reply:

Corrected.

  1. Methods section:

I could not clearly understand the participant selection of the study. There were initially 238 124, 100 201, 247 464, and 188 622 participants recruited in the series of surveys from 1982 to 2012, respectively. Data were available for 39 084, 58 316, 52 426, and 55 051 participants, respectively. In each year, 16% (39084/238124), 58% (58316/100201), 21% (52426/247464), and 29% (55051/188622) of participants were included in the analysis. Is this correct? Why the percentages of participant selection largely differ? I feel most of the participants of the surveys excluded from the analysis. Why? I suggest showing participant selection using a brief figure.

 Reply:

More detailed information was added in to the Methods Part “2.1 Study population and sampling“ as follows:

 All CNNS rounds collected identical data from household and dietary interviews, body measurements and laboratory tests. Some participants were selected to participate in certain survey item while others participate in another.

  1. Methods section:

What was the definition of urban or rural areas? Is this just based on the population?

  Reply:

Urban and rural areas was defined by the Statistical Bureau and used national wide in China.

  1. Results section (Table 1):

It will become clearer to know population aging and change in education levels if the authors add means (or medians) of the variables in Table 1.

  Reply:

Though the age distribution was changing across the survey rounds, we had adjusted the dietary intake representing the intake of a fixed population. In order to compare the dietary intake across years, the weights were derived from the sampling probability of the 2010 Chinese population aged 20 years or older (based on census data) and applied to estimate the representative dietary intake in each survey round.

The demographic information in Table 1 only described the characteristics of the participants. All the results in other tables were weighed.

  1. Results section (Table 2):

Table 2 is not totally shown. Table 2 should be simple to delete 1992 vs …. (all columns to the right of “p for trend”) and “Range” and “subgroups” lines. I can see these results described in the main text.

   Reply:

The differences between years were used to describe the sharp rising or moderate rising trends of energy and macronutrient composition in population.

  1. Results section (Table 3-5):

Same as comments for Table 2 above, the columns “Difference between Rounds, % (95%CI)” seems to be unnecessary. These results may describe in the main text only, if necessary. I think it is not unnecessary showing “Range”, lines as well.

    Reply:

The differences between years were used to describe the sharp rising or moderate rising trends of energy and macronutrient composition in population.

The “Range” explained the disparity degree within the groups.

  1. Results section (description for Table 3):

Lines 126-127: ~except the subgroup of those educated over 15 years.

I could not understand why except those educated over 15 years because of trend p <0.01.

  Reply:

The trend of the estimated percentage of energy intake from fat among the well-educated population was rising at the former survey rounds but declined from 37.4% to 36.6% during 2002 to 2012 (2012 vs 2002 difference, -0.8%; 95% CI -1.6% to 0.0%). While the trends of the other two macronutrients were rising all the way among other subgroups. This was the difference.

  1. Discussion section:

Line 20:

“NCD” should be expanded, followed by the abbreviation. This is the first use in the manuscript.

 Reply:

Corrected.